# The Proteome of Acute Muscle Pain: Observations from Acute Hypertonic-Saline-Induced Pain in Humans

**DOI:** 10.3390/ijms262411922

**Published:** 2025-12-10

**Authors:** Pauline Jubin, Marie Amigo, Daniel Boulton, David A. Mahns, Saad S. Nagi, James S. Dunn

**Affiliations:** 1School of Medicine, Western Sydney University, Campbelltown, NSW 2560, Australia; p.jubin@westernsydney.edu.au (P.J.); m.amigo@westernsydney.edu.au (M.A.); daniel.boulton@westernsydney.edu.au (D.B.); d.mahns@westernsydney.edu.au (D.A.M.); james.dunn@westernsydney.edu.au (J.S.D.); 2Department of Biomedical and Clinical Sciences, Linköping University, 582 25 Linköping, Sweden

**Keywords:** pain mechanisms, muscle pain, nociception, molecular signaling, acute pain proteome, allodynia, mechanical hyperalgesia

## Abstract

Despite the widespread use of experimental acute pain models, little exploration has been undertaken on the acute pain proteome in humans. We resolved to explore molecular alterations evoked by hypertonic saline (HS)-induced acute muscle pain and to map the spread of mechanical hyperalgesia. This study used a two-cohort design in healthy participants. Cohort one (*n* = 16) underwent intermittent blood sampling prior to, during, and following intramuscular HS (5%) infusion to allow for the discovery of the proteomic and cytokine profile of acute muscle pain. Cohort two (*n* = 10) underwent bilateral sensory testing during HS infusion, to map the spread of mechanical hyperalgesia. Molecular analysis in cohort one revealed a broad array of proteins and cytokines showing altered expression in response to acute muscle pain. Particularly, these alterations were linked to metabolism and immune response pathways suggestive of systemic effects of acute pain. Cohort two revealed a significant mechanical hyperalgesia which emerged in a distributed pattern over the ipsilateral limb to HS infusion. However, despite systemic molecular alterations, no such mechanical hyperalgesia was observed in the contralateral limb. This study demonstrates systemic molecular alterations resultant from acute HS-induced muscle pain, accompanied by spatially constrained sensory interactions. This dissociation implies that, at least in acute sensitization, widespread molecular changes may not necessarily translate into a correspondingly widespread sensory phenotype.

## 1. Introduction

Intramuscular administration of hypertonic saline (HS) has been used for decades to reliably evoke a clinically relevant pain in healthy human participants [1,2]. Intramuscular HS results in reports of aching and cramping sensations in addition to a referral of pain away from the site of HS administration [3,4]. The referral of pain from HS can often be felt in neighboring and adjacent tissues including muscle, joint, and skin [2,5,6,7,8,9,10,11]. Additionally, longitudinal observations of local and referred pain regions in response to repeat HS infusions have shown that the pain response undergoes a spatiotemporal shift with a decrease in the spread of local pain but an increase in the perception of referred pain when infusions of HS are administered at weekly intervals [12].

Our lab has previously shown that repeat injections of HS (5%) into the tibialis anterior (TA) muscle of the leg results in a profound and bilateral mechanical hyperalgesia (increased pain to normally painful stimuli) as measured by bilateral pressure pain threshold (PPT) testing performed across the TA muscles [11]. This is in addition to the emergence of a bilateral cold allodynia, suggesting that repeated episodes of unilateral muscle pain induced by HS administration can produce central sensitization resulting in mirror-image-like sensory disturbances in the contralateral limb. It remains unknown whether a single (one-off) HS infusion can produce a contralateral mechanical hyperalgesia. Indeed, we have shown that acute HS infusions can evoke a contralateral and remote hyperalgesia response in response to sub-perceptual stimulation of muscle afferents [5], but it remains to be seen whether a more overt stimulus such as PPT testing can evoke this same result.

Whilst PPT testing can provide insight into the sensory impacts of muscle pain, the underlying molecular mechanisms of altered sensitivity in both acute and prolonged time courses remain unclear. Proteomics and whole blood analysis provide a promising approach which permits the identification and quantification of the pain proteome. Thus far, proteomics in pain research has been heavily skewed towards examination of chronic pain pathologies such as chronic widespread pain (CWP) [13,14], fibromyalgia [15,16], migraine [17], and chronic low back pain [18]. These examinations have uncovered key differences between the proteomes of healthy controls and patients with chronic pain conditions, only leading to further questions about the processes which see a ‘healthy person’ transition into chronic pain over time. Notably, proteomic studies tracking acute changes in response to transient bouts of pain are lacking and could provide valuable insight into the processes initiating prolonged pain responses and pathologies. This is particularly pertinent given that acute musculoskeletal pain is a universally known sensation felt by almost the entirety of the human population at some point in time.

In the current study, we aimed to investigate the molecular underpinnings of the bilateral pain interactions we previously reported [5] and further assess the sensory modalities altered in acute muscle pain. Two cohorts were recruited for this study; one which underwent HS infusion accompanied with intermittent blood sampling to evaluate the molecular changes, and a second cohort which undertook bilateral PPT testing to assess whether acute pain could evoke a bilateral mechanical hyperalgesia as seen in our prolonged experimental pain model [11].

## 2. Results

### 2.1. Cohort One—Molecular Data (Proteomics and Cytokines)

All 16 participants recruited for this stage of the study were able to achieve a stable moderate baseline pain between 4 and 6 out of 10 in response to intramuscular 5% HS infusion. Of these 16 participants, successful venepuncture was achieved in 13 individuals. For each of the 13 samples at each time point (T0, T10, T20 and T30), technical duplicates were processed.

Untargeted plasma-based proteomics identified 1671 quantifiable proteins. By applying filters as per the methods (confidence score ≥50, ≥1 unique peptides, and ≤30 ppm mass error), 547 proteins were identified for further statistical analysis. Analysis of the proteomics and targeted cytokine/chemokine analysis revealed 29 proteins (Table 1) and 2 cytokines (Figure 1) differentially expressed between T0 and T10. Furthermore, comparisons between T0 and T20 revealed differential expression of 17 proteins (Table 1) and 5 cytokines (Figure 1), and 12 proteins were differentially expressed between T0 and recovery.

Molecular changes were observed in response to acute muscle pain, mostly relating to metabolism, immunity and coagulation, protein regulation tissue structure, and vesicle trafficking (Figure 2). A greater number of differentially expressed proteins were observed at the T10 time point (pain onset) when compared to the T20 time point (stable pain plateau), potentially indicating a role of these proteins in dynamic pain or pain maintenance and tolerance. Expression differences were also observed for 12 proteins at our recovery time point when compared to baseline (T0), of these 4 proteins were also identified at T10 but not T20, suggesting that they may be involved in the dynamic phases of pain perception. Cytokine analysis revealed increased recruitment of leukocytes and, subsequently, sustained markers of inflammation. All cytokines that showed significantly altered expression at T10 were also significantly altered at T20 and no differences were found between T10 and T20 (Figure 1).

### 2.2. Cohort Two—Assessment of Mechanical Hyperalgesia

A stable moderate baseline pain (between 4 and 6 out of 10 on the pain scale) was achieved in all participants in cohort two (*n* = 10). A two-way RM ANOVA found a significant effect of treatment condition (baseline or HS infusion) as well as significant differences between the results obtained at different applied forces (*p* < 0.0001). The generated heat maps of mechanical hypersensitivity reveal a lowering of mechanical thresholds in the regions directly adjacent to the HS infusion site, with sites that were ~1 cm away from the infusion needle reporting a PPT in almost all participants (Figure 3).

In all participants, a significant mechanical hyperalgesia was observed in the ipsilateral leg during HS infusion with an average of 8.7 ± 1.4 points reported as painful at 30 N compared to 2.7 ± 0.6 tender points at baseline (*p* < 0.0001) (Table 2 and Figure 4). This significant increase between baseline and HS infusion pain conditions was observed for forces of 15 N and higher with testing at 5 N and 10 N producing less than one tender point on average under both conditions with no significant variation (*p* > 0.05).

In contrast to the ipsilateral limb, the contralateral limb showed no significant change between baseline and HS infusion conditions for any level of force application (Figure 4). At the maximum applied force of 30 N, participants reported an average of 1.8 ± 0.7 tender points at baseline and 1.7 ± 0.6 during the HS infusion (*p* = 0.9988). Additionally, the mean number of tender points at 30 N between the ipsilateral and contralateral limbs were significantly different in the HS infusion condition, with the contralateral HS infusion condition being no different from either the contralateral or ipsilateral baselines.

#### Minimum Pressure Pain Thresholds

When the minimum applied force at which a participant reported pain was examined, a significant variation between the baseline and HS infusion conditions again emerged in the data for the ipsilateral limb (*p* < 0.05, F = 10.72).

At baseline on the ipsilateral leg, 9 of the 10 participants reported a PPT at one of the 30 test sites within the testable range (≤30 N), whilst only 1 participant did not report any sites as tender in response to this force application (Figure 5). Of the 9 who did report a PPT, the mean threshold was 22.22 ± 2.2 N. During HS infusion, all 10 participants reported a PPT within the testable range (≤30 N) with a significantly reduced mean minimum threshold of 13.5 ± 1.7 N (*p* = 0.0402).

In the contralateral limb, only 5 participants reported a PPT within the testable range during at baseline with a mean threshold of 22.0 ± 3.4 N. In the HS infusion condition, a further participant reported a PPT within the testable range (taking the tally to 6); however, no significant variation from baseline emerged in the mean minimum PPT for the contralateral leg (20.8 ± 3.3 N; *p* = 0.9914).

## 3. Discussion

This study aimed to elucidate the proteomic underpinnings of previously observed HS-induced muscle pain interactions and clarify whether the bilateral mechanical hyperalgesia observed in prolonged experimentally induced muscle pain models can be evoked in an acute setting. Informed by the interactions we had previously observed [5], we hypothesized that the requisite molecular underpinnings for diffuse mechanical hyperalgesia and hypersensitivity would be sufficiently altered even in response to a relatively short bout (~20 min) of muscle pain. Indeed, we were able to detect significant molecular changes in participants, suggestive of such changes. However, given we had previously reported a bilateral sensitivity to a normally innocuous stimulus (normal saline infusion) [5], and mechanical hyperalgesia following repeated injections of HS [11], we were surprised that we were unable to detect the bilateral interaction observed via mechanical algometry.

Significantly, we believe this is the first direct report that an acute HS-induced muscle pain is sufficient to drive systemic molecular alterations detectable in the systemic circulation within 10–20 min. Importantly, pathways identified in our acute pain model proteomic analysis, such as complement and coagulation, regulation of actin cytoskeleton, metabolic processes and post-translational modifications have been identified as linked to chronic pain prognosis [18]. Notably, we have seen direct overlap in pathways and even individual proteins with chronic conditions including chronic migraine [17], fibromyalgia [14,20], and rheumatoid arthritis [21].

Infusion of 5% HS in muscle tissues causes tissue and cell depolarization via voltage-gated channels activation. Sodium-gated channels, notably Na_v_1.9 and Na_v_1.8, and transient receptor potential (TRP) channels, such as TRPV1 and TRPA1, are known to participate in pain and sensitization [22]. The activation of these channels excites nociceptive nerve terminals and induce calcium influx resulting in depolarisation/sensitisation of the nerve terminal [23].

In our study we found a decreased abundance of most hemoglobin (Hb) subunits (HBA1, HBA2, HBB, HBD, and HBE1). Lower levels of these subunits may suggest increased levels of free Hb and reduced Hb-bound iron, which could promote oxidative stress and the generation of reactive oxygen species (ROS). In what could be seen as a compensatory measure of this, we also observed increased expression of HP, consistent with an inflammatory response, as HP binds free Hb to mitigate heme- and iron-driven ROS production [24]. In parallel, we detected a lower abundance of LTF, a protein with iron sequestration activity [25]. This decrease may indicate impaired iron binding, leading to elevated levels of labile iron, thereby enhancing oxidative stress and tissue inflammation. Such effects may also contribute to lipid and protein oxidation, potentially reflected by the reduced abundance of ApoB, as oxidative modifications can alter the mass-to-charge (m/z) ratio and consequently affect proteomic detection [26]. Additionally, LTF is primarily released by neutrophils [27], whereas HP can associate with neutrophils, particularly through their involvement in hemoglobin scavenging [24]. The observed difference in LTF and HP expression may therefore point to a limited or altered neutrophil response, characterized by predominant Hb scavenging activity (via HP) but reduced free-iron sequestration (via LTF). Altogether, these findings suggest a potential increase in free iron, leading to enhanced oxidative stress, tissue inflammation, and ROS production. Emerging evidence further indicates that ROS can contribute to peripheral sensitization by interacting with TRP channels and modulating the expression of pro-inflammatory cytokines [28].

Our molecular analysis also revealed markers of increased apoptosis following HS infusion. There was a significant increase in the expression of TRAIL (TNFSF10 cytokine), responsible for cell apoptosis, potentially acting via the activation of caspase 3, a pathway known to be involved in pain [29,30]. Decreased FLT3LG might lead to less functional dendritic cells and fewer T-cell activation [31], which could in turn lead to a reduction in IL17C since this cytokine is activated T-cell derived [32]. Decreased IL17C, together with decreased CCL2 (responsible for recruitment of monocytes, memory T-cells and dendritic cells [33]), suggest dynamic changes in immune cell recruitment and activation. This may be indicative of a disrupted immune response and tissue homeostasis. These observations could be most pertinent to the dynamic/early stages of pain, with all participants reporting a sustained pain at T20, possibly explained by a dynamic rebalancing of immune clearance and recruitment.

In conjunction with a decreased expression of CCL19, we also observed downregulation of IGHV, both of which are involved in the adaptive immune response [34,35]. These dynamic variations in the adaptive immune response are also evidenced by downregulation of AP3B2 and PAM [36,37], involved in protein trafficking to lysosome-related vesicles and peptide processing during secretion, respectively, possibly indicating a dynamic alteration of antigen processing and presentation activity.

Immunity is also implicated by complement and coagulation cascades. In our study we observed that decreased expression of CFI, an inhibitor of the complement cascade [38], was accompanied by increased expression of C3b and C4b. C3b and C4b are necessary to cleave C5 into C5a and C5b, with C5a being a known agent/protein that can sensitize nociceptors, leading to opening of TRP channels [39]. In parallel, during pain onset and establishment (T10) there was notable downregulation of F12, an important component of the intrinsic coagulation pathway. Additionally, F12 is pivotal in the kallikrein pathway and subsequent bradykinin production [40], with lower F12 levels during pain onset leading to a possible reduction in bradykinin production and a blunting of peripheral sensitization, in an apparent protective attempt to reduce the immediate inflammatory pain. However, this may be countered by the concurrent increase in AGT, the precursor of angiotensins [41], resulting in a pro-inflammatory response.

We observed potential disruption of chromatin structure, via the downregulation of proteins associated with cohesion complex and chromatin regulation (SMC3, CHD2, ATF7IP, GTF3C1, RBMX). This alteration in chromatin structure might cause altered transcription, potentially dysregulating apoptosis, stress response, immune signaling [42] and pain sensitivity via ion channel expression [43].

We also observed an upregulation of proteins involved in GTPases regulation (SYTL2, RANBP3, PLXNC1, RABEP1). Small GTPases are essential for cytoskeleton dynamics, cell polarity and migration, including immune cells [44], as well as in sensory neuron development inclusive of nociceptors [45]. Under inflammatory conditions, it has been shown that small GTPases can regulate leukocyte recruitment and function [46]. In parallel, we observed a downregulation of RAI14, an actin-associated protein acting downstream of GTPases, and an upregulation of ITIH1, a protein that stabilizes the ECM, as well as un downregulation of CLIP1 (microtubule stability) and upregulation of MAST1 (microtubule and cytoskeletal dynamics signaling/modulation). Together, these changes suggest cytoskeletal and ECM remodeling that could affect cell migration, trafficking, and inflammatory response, as well as impacting the activation of mechanosensory channels [47], further enhancing pain sensitivity.

Altogether, these results suggest an impaired adaptive and anti-inflammatory response following an acute bout of musculoskeletal pain. The vast majority of these molecular alterations resolve following the cessation of HS-induced muscle pain, suggesting a causal link between pain induction and a transiently altered immune profile. However, as this represents the first examination of proteomics in an acute experimental pain setting, and due to the comparatively small sample size for proteomics, mechanistic and pathway implications need to be explored further and with larger datasets. Future expansion of the acute pain proteome will bring greater certainty to the molecular pathways which underpin acute pain sensation in humans. Additionally, more detailed insight into the temporal dynamics of cytokine changes in relation to pain resolution (T30) would be beneficial in future work, as we did not have the resources to explore this within this study.

Intriguingly, despite the multitude of molecular alterations observed in response to acute muscle pain, there was a marked ipsilateral constraint to the evoked mechanical hyperalgesia. Cohort two of this study utilized a 30-point grid overlying the site of HS infusion, allowing for the detailed mapping of the regional distribution of HS-evoked mechanical hyperalgesia, not limited to the HS-infused TA muscle, but also in additional regions within the same and adjacent muscle compartments. The HS-infused muscle in cohort two did differ from the site of pain in cohort one (FCU), yet our data showed that 5% HS evoked a similar pain intensity and identical pain qualities. The move to the TA and anterolateral leg compartment facilitated a broader collection area for PPTs and a more thorough assessment of mechanical hyperalgesia and thresholds in the context of acute muscle pain. Yet, cohort two revealed that a single infusion of HS into the TA muscle was insufficient to evoke a bilateral mechanical hyperalgesia, suggesting that repeated noxious stimulation of the periphery, likely over days [11], is required to produce the requisite changes at the central level for a bilateral effect to this stimulus to be observed. Whilst that earlier study observed a comparable number of tender points at baseline as the current work, the cumulative tender points evoked at peak hypersensitivity (following repeated HS bolus injections) are markedly different, with many more test sites reported as painful in response to repeated HS exposure as opposed to our singular acute infusion. Furthermore, that study was not conducted in the presence of an ongoing overt baseline pain with repeated HS injections resulting in a largely sub-perceptual hypersensitivity which required external stimulus to emerge. This is unlike the current study where the acute perception of ongoing moderate musculoskeletal pain may have acted as a confounding factor, potentially accounting for the lack of contralateral effect.

In both cohorts, HS was administered for at least 10 min prior to follow-up testing (either blood sampling or PPT) in order to establish a stable, moderate baseline pain. This period of nociceptive input could have allowed for sensitization of wide-dynamic range neurons in the dorsal horn [48,49] and thus a state of central sensitization to develop. Indeed, the clinical correlates of central sensitization [49,50] are apparent in the HS infusion pain model with muscle hyperalgesia and cutaneous allodynia reported in this and previous works [5,9,10,11,51]. Nonetheless, in the present study, it is not possible to determine whether the observed hyperalgesia reflects central sensitization, peripheral sensitization, or both. The presence of hyperalgesia may involve contributions from both peripheral and central mechanisms, while the absence of a mirror image effect does not, in itself, rule out central involvement.

Our group has shown that acute muscle pain evoked by HS infusion can be modulated (increased) by concurrent infusion of normal saline (NS, 0.9%) in adjacent, contralateral, and remote muscles [5]. NS infusion is imperceptible under normal conditions, and during concurrent NS-HS infusion, the increase in pain was almost always reported at the site of HS infusion and not the NS infusion site. In contrast, in the current study, we were studying the somatotopic spread of mechanical hyperalgesia (i.e., reduction in PPT) and found no evidence of contralateral effects. This raises the question of whether stimulus modality alone explains the differential spread, or whether distinct underlying mechanisms drive hyperalgesia (as in this study) versus allodynia [5]. The former is locally constrained initially, requiring repeated provocations over time to show bilateral spread, while the latter reflects a rapid, widespread facilitation by low-threshold muscle inputs. Thus, these phenomena seem to be operating on different timescales, and the stimulus properties together with the frequency of provocations, determine which mechanism is engaged.

Overall, we were able to establish that an acute intramuscular HS infusion results in a diffuse muscle pain that, whilst constrained to the ipsilateral limb, is sufficient to evoke significant molecular alterations. Proteomics analysis revealed differential expression of proteins after as little as 10 min of muscle pain, with some of the identified proteins also observed in chronic pain conditions. The dysregulation of these proteins in acute and more prolonged settings is an area that warrants further investigation to understand the molecular mechanisms of pain and unlock the pain proteome which may facilitate the transition from acute to chronic pain.

## 4. Materials and Methods

Naïve healthy participants with no reported history of musculoskeletal or neurological disorders were recruited for both arms of this study. Cohort 1 consisted of 16 participants (8 females) and cohort 2 consisted of 10 individuals (3 females). Participants were naïve in that they were blinded to the desired outcomes of each study. Participants were also asked to abstain from intense physical activity for the 48 h preceding their involvement in the study, as such activity is known to produce muscle hypersensitivity [11,12,52]. Informed written consent was obtained from all participants prior to each experimental session. The study was approved by the Human Research Ethics Committee of Western Sydney University (approval numbers H9190 and H13204) in accordance with the Revised Declaration of Helsinki.

### 4.1. Cohort 1—Proteomic Changes Asscoiated with Acute Pain

Our first cohort underwent HS infusion with concurrent periodic blood collection to ascertain proteomic and cytokine changes associated with sustained experimentally induced muscle pain. For this cohort (*n* = 16), the 5% HS (AstraZeneca Pty Ltd., North Ryde, New South Wales, Australia) infusion was administered to the mid-point of the belly of the flexor carpi ulnaris (FCU) muscle of the forearm. Either left or right arm was chosen in a randomized manner with no preference in the methods for dominant or non-dominant arm. The HS infusion was delivered at a variable rate (50–250 µL/min) to achieve a moderate baseline pain in individuals (rated 4–6 out of 10). The infusion protocol followed the methods in our prior published works [5,9,10,11].

Pain ratings were continuously recorded using the ADInstruments Response Meter connected to the ADInstruments Powerlab (ADInstruments, Dunedin, New Zealand). The Response Meter had a slide control, and the pain scale was divided into 10 equal segments within a range of 0 (no pain) to 10 (worst pain). The participants were free to rate their pain as they felt throughout the duration of the study and remained in control of the VAS at all times. The infusion in this cohort was subject to strict time controls with the infusion lasting 20 min total in this group.

#### 4.1.1. Blood Sample Collection and Plasma Isolation

Venous blood was collected from the antecubital region of participants prior to the commencement of the HS infusion (T0—baseline). This would then be compared to blood collected after 10 min of infusion (T10—pain onset), 20 min (T20—stable pain plateau). After the T20 blood collection the HS infusion was ceased and pain allowed to return to 0. Once the participant no longer reported pain, blood was collected once more to serve as a ‘recovery’ sample. At each time point, 9 mL of blood was collected in K2-EDTA coated tubes (BD Livingstone, Mascot, NSW, Australia, #BD367525).

A balanced salt solution (BSS) was previously prepared by mixing solution A (Anhydrous D-glucose 1%, CaCl_2_ × 2H_2_O 0.0074 g/L, MgCl_2_ × 6H_2_O 0.1992 g/L, KCl 0.4026 g/L, TRIS 17.565 g/L, pH 7.6) with solution B (NaCl 8.19 g/L) at a ratio of 1:9. Whole blood was diluted with this BSS (*v*/*v*) before being carefully layered over Ficoll Plaque^®^ Plus (Cytiva, Marlborough, MA, USA, #17144002) (3:4) and centrifuged for 40 min at 400× *g* to isolate plasma.

#### 4.1.2. Proteomics

To increase protein recovery, plasma was spun at 18,000× *g* for 30 min at 4 °C to separate cytosolic proteins and membrane proteins. Protein estimation was undertaken using NanoDrop Lite spectrophotometer (Thermo Fisher Scientific, Waltham, MA, USA) and was resuspended at the same concentration for all samples in 50 mM ammonium bicarbonate. Reduction and alkylation of the proteins were performed using dithiothreitol (DTT, Roche, Basel, Switzerland, #10197777001) for 30 min in the dark at 60 °C, and iodoacetamide (IAA, Sigma-Aldrich, St Louis, MO, USA, #I1149) for 30 min in the dark at room temperature, respectively, then trypsin (Promega, Madison, WI, USA, #V5280) was used for enzymatic digestion overnight at 4 °C to yield peptides. Trypsin was stopped using formic acid and cytosolic samples were cleaned using solid phase extraction before all samples were vacuum-dried and resuspended in 0.1% formic acid at 1.6 µg/µL. Samples were transferred into total recovery vials (Thermo Fisher Scientific, #11093563), and 0.5 µL was injected for analysis on a Waters nanoAcquity UPLC sample manager coupled with a Waters Synapt G2-Si QToF instrument (Waters Corporation, Milford, MA, USA) fitted with an ESI source.

All samples were run in a single batch and as duplicates to quantify variability and reproducibility. All samples were imported into Waters Progenesis QI for proteomics (Waters Corporation, v.4.1.6675.48614) to analyze the obtained peptides and retrieved proteins, with chromatograms aligning >87% to a pooled sample (quality control sample). Peptides and proteins identification was performed on Progenesis using Ion Accounting identification method searching the human proteome database from UniProt [53] FASTA file. Obtained proteins were filtered using mass error (≤30 ppm) with at least one unique peptide and confidence score (≥50) output by the Progenesis software to control for false positive results. To assess the correlation between duplicate, samples were submitted to Spearman test, and all replicates had a score of >0.9, showing great consistency between replicates and a negligeable batch effect. This was replicated for each fraction.

The mass spectrometry proteomics data have been deposited to the ProteomeXchange Consortium via the PRIDE [54] partner repository with the dataset identifier PXD070131.

#### 4.1.3. Cytokine Analysis

Cytokines were analyzed using an Olink Target 48 Cytokine plate following the protocol provided by the manufacturer (Olink, Uppsala, Sweden) to allow for the absolute quantification of 45 targeted cytokines. Plasma samples were incubated overnight with antibody pairs with attached DNA tags, to allow for the specific recognition of DNA tagged proteins. Target protein–DNA tags pairs were hybridized to allow for the specific binding then extended to generate unique DNA reporter sequences to each target protein. Unique DNA reporters were amplified using regular polymerase chain reaction (PCR), then quantified using high throughput real-time quantitative PCR (qPCR) on the Olink Signature Q100 system, with data analyzed using the NPX^TM^ Signature software (v.2.0.2).

#### 4.1.4. Identification, Enrichment, and Pathway Analysis

When comparing T0, T10, T20, and recovery, the corresponding gene names of obtained proteins were recovered from the protein accession number using UnitProtKB database [53]. The gene list was submitted to Metascape [55], STRING [19], KEGG [56], and Reactome [57]. These platforms allowed for Gene Ontology (GO) enrichment, pathways and protein–protein interactions (PPIs) analysis. Protein functions were derived from GeneCards [58].

### 4.2. Cohort 2—Examination of Mechanical Hyperalgesia During Acute Muscle Pain

With participants comfortably seated in a semi-reclined position in a chair, a 4 × 18 cm grid—consisting of 30 spots in a 3 (columns) by 10 (rows) arrangement with a 2 cm distance between each grid point—was drawn over the anterolateral compartment of both legs with the grid centered around the belly of the TA muscle of the leg, as has been used for PPT testing in previous studies [11,51,59].

#### 4.2.1. Pressure Pain Threshold (PPT) Testing

Mechanical algometry was performed bilaterally at two time points to assess PPTs for each individual location on the 30-point grid overlying the TA muscle. The time points tested were prior to HS infusion (baseline) and during HS infusion (steady pain state).

A force of 30 N was applied to each individual site (*n* = 30 per leg—i.e., 60 sites per participant per condition) for ~3 s using a 1 cm^2^ rubber tip pressure gauge (Pain Test Model FPN 50 Algometer, Wagner Instruments, Greenwich, Connecticut, USA). If pain was reported by a participant, then a progressively smaller force (in decrements of 5 N) was applied until the participant no longer reported the stimulation as painful. An inter-stimulus interval of 10 s was used, and threshold values were confirmed by testing adjacent and remote sites in a pseudo-random order before re-testing the site reported as tender/painful at a lower force. The upper end of the PPT test range (30 N) was selected so as not to evoke a painful response in more than ~30% of the population and aligns with reports that 30 N is the approximate upper border of the innocuous pressure range [60].

#### 4.2.2. Hypertonic Saline (HS) Infusion

After mechanical algometry was performed, and PPTs obtained for all 60 sites across both legs at baseline, in the same experimental sitting, an infusion of 5% HS (AstraZeneca Pty Ltd., North Ryde, New South Wales, Australia) was commenced in the belly of the TA muscle of the leg. Either the left or right leg was chosen in a randomized manner with no preference in the methods for dominant or non-dominant leg. The site of infusion was at approximately the mid-point of the 30-point grid, approximately 1 cm away from the test sites in the middle column of the 5th and 6th rows.

The infusion rate varied between participants with a range of 50–250 µL/min depending on the pain rating of the participant. Once the HS infusion elicited a moderate baseline pain of between 4 and 6 on the pain scale, the infusion was continued for a further 5–10 min to allow for stability of the baseline pain. Following this, with the HS infusion still continuing, PPT testing was repeated bilaterally to observe the potential bilateral changes in muscle hyperalgesia as a result of acute HS-evoked muscle pain. At the completion of bilateral PPT testing, the HS infusion was stopped, and the participant monitored for the next 5–10 min until the HS-evoked pain had subsided.

### 4.3. Statistical Analysis

#### 4.3.1. Molecular Data

Proteins and cytokines raw abundances were normalized on T0 and obtained fold changes were used for further analyses. Comparison of our time points (T0—baseline, T10—10 min of infusion, T20—20 min of infusion, and recovery) was undertaken using GraphPad Prism (v.10.6.1).

Distribution of the data was analyzed using D’Agostino and Pearson omnibus normality test and re-examined using Shapiro–Wilk normality tests. Once the data was confirmed to not conform to a normal distribution, we applied a two-way ANOVA fitting mixed effects (correcting for missing values). As participants were identical for all the different time points (T0, T10, T20, and recovery), we applied matched values to the dataset and examined multiple comparisons with a simple row effect to compare proteins independently. Dunnett post hoc test was applied to the ANOVA results to control for Type I and family-wise error as it is a powerful and efficient test focusing on the comparison of several groups to a control.

Cytokine analysis was undertaken using the same statistical methodology as identified proteins.

#### 4.3.2. Sensory Data

The analysis of the data was undertaken with a two-way repeated measures (RM) ANOVA to compare the PPTs obtained bilaterally at baseline (no HS pain) and HS-induced pain conditions across each of the tested grid points (*n* = 30 per limb, per condition, per participant). Tukey’s multiple comparison test was performed to identify individual changes.

Additionally, an analysis of the minimum PPTs was performed using a one-way ANOVA with Tukey’s multiple comparison test to compare test sites and conditions. D’Agostino and Pearson omnibus was performed to confirm the presence of normally distributed data. PPT data are presented as average ± standard error of the mean (SEM). The regional distribution of PPT was also examined and is presented as ‘heat maps’.

## 5. Conclusions

An acute bout of HS-induced muscle pain in humans is sufficient to evoke molecular alterations with accompanying mechanical hyperalgesia. This is the first exploration of the proteomic alterations induced by acute experimentally induced pain, giving insights into the molecular pathways of pain in humans. Several of the proteins which showed differential expression during acute pain have also been implicated in chronic pain conditions. We hypothesize that the molecular overlap between acute pain and chronic pain conditions may help shed mechanistic insight on the transition from acute to chronic pain.

## Figures and Tables

**Figure 1 ijms-26-11922-f001:**
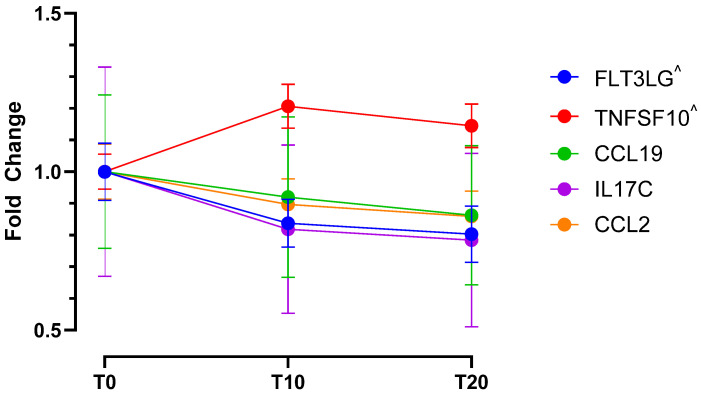
**Temporal dynamics of significantly altered cytokine expression in response to acute HS-induced pain.** Differentially expressed cytokines and their max fold changes compared to baseline (T0), during pain establishment (T10), and during sustained acute muscle pain (T20). All cytokines show significantly altered expression at the T20 time point, cytokines marked with ^ show additional differential expression at T10 compared to baseline (T0).

**Figure 2 ijms-26-11922-f002:**
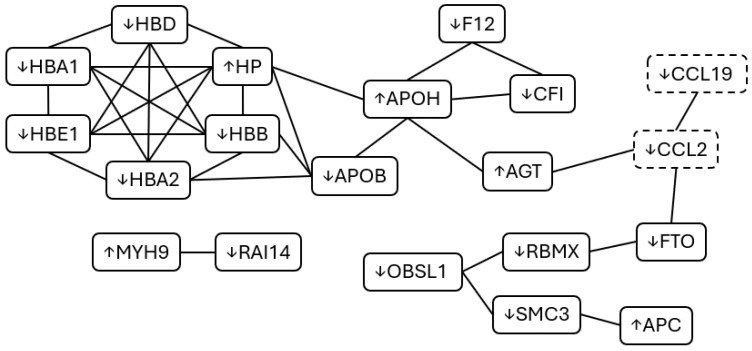
**Protein–Protein Interaction (PPI) of significantly altered proteins and cytokines during acute muscle pain.** Several interrelated proteins were identified as differentially expressed in response to acute muscle pain with some variation between our different blood sampling time points. Differentially expressed cytokines are also displayed for each of these time points (dashed boxes). Whether expression increased or decreased is represented by arrows next to the protein/cytokine name. Adapted from STRING [19].

**Figure 3 ijms-26-11922-f003:**
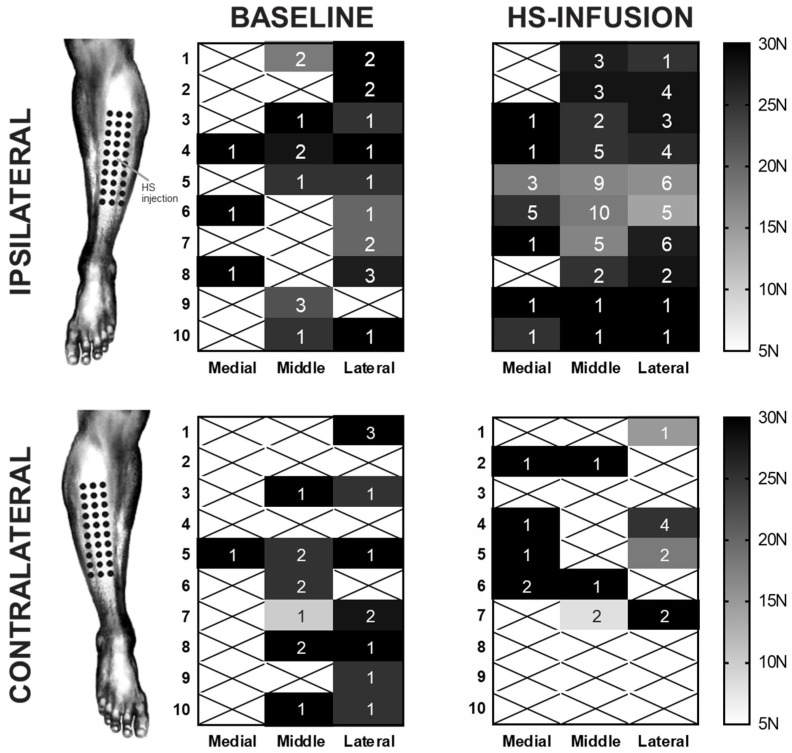
**The somatotopic spread of hypertonic-saline-induced mechanical hyperalgesia in the ipsilateral and contralateral limbs**. The mean observed pressure pain thresholds for each of 30 sites across the anterolateral leg are displayed with the n for each site (out of the possible 10 subjects) displayed. Sites where no PPT was reported are marked with an X. During the HS infusion, a marked increase in mechanical hypersensitivity occurred directly around the region of the infusion site which was between the fifth and sixth rows in the middle column. Concurrent HS infusion appeared to have no impact on the mechanical sensitivity of the contralateral limb.

**Figure 4 ijms-26-11922-f004:**
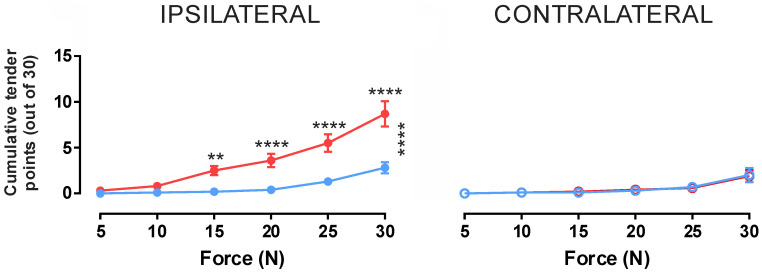
**Cumulative tender points at varying forces at bilateral testing sites during hypertonic saline infusion into the tibialis anterior muscle.** Blue indicates values taken prior to HS infusion whilst red indicates values obtained during HS infusion. Additionally, vertical asterisks (*) indicates significant variation using 2-way ANOVA, whilst horizontal asterisks indicate significance using post hoc Tukey analysis, with *p* ≤ 0.05 *, *p* ≤ 0.01 **, *p* ≤ 0.001 ***, *p* ≤ 0.0001 ****. For applied forces of 15 N and above, a significant effect of HS infusion was observed in the ipsilateral limb (*p* > 0.05); however, there was no effect in the contralateral limb, suggesting an ipsilaterally constrained mechanical hypersensitivity evoked by intramuscular HS infusion.

**Figure 5 ijms-26-11922-f005:**
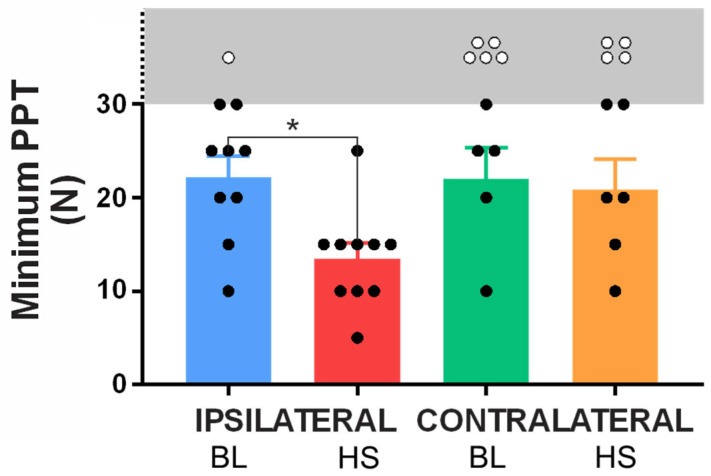
**Minimum pressure pain thresholds (PPTs) in baseline (BL) and HS infusion conditions for the ipsilateral and contralateral limbs.** Subjects who reported a minimum PPT within the testable range (5–30 N) are marked by a black circle, whilst subjects who did not exhibit a threshold within the testing parameters are marked with a white circle (exact threshold not tested). The gray region and broken *Y*-axis demark thresholds above our testing range. A significant decrease (* *p* < 0.05) in PPT was observed in the ipsilateral limb between the baseline and HS infusion conditions, however no such interaction was observed in the contralateral limb. n= 10.

**Table 1 ijms-26-11922-t001:** Differentially expressed proteins and their max fold change compared to baseline (T0) during pain establishment (T10), sustained acute muscle pain (T20), and recovery (R). Significantly differentially expressed proteins are bold, with *p* ≤ 0.05 *, *p* ≤ 0.01 **, *p* ≤ 0.001 ***, *p* ≤ 0.0001 ****, non-significant differences are shown in italics, # no data.

Functional Group	Protein	Fold Change		
		T10	T20	T30
Gene expression regulation	APC Adenomatous Polyposis Coli	*1.162* *± 0.100*	*1.156* *± 0.090*	**1.248 *** **± 0.073**
	ATF7IP Activating Transcription Factor 7 Interacting Protein	**0.859 *** **± 0.062**	*0.967* *± 0.047*	*0.995* *± 0.051*
	CHD2 Chromodomain Helicase DNA Binding Protein 2	**0.883 *** **± 0.049**	*0.939* *± 0.030*	*0.939* *± 0.029*
	CNTLN Centlein	**0.779 *** **± 0.069**	*0.784* *± 0.069*	*0.909* *± 0.006*
	DDX42 DEAD-Box Helicase 42	**0.741 ***** **± 0.038**	**0.845 *** **± 0.040**	*0.831* *± 0.043*
	ERCC6L2 ERCC Excision Repair 6 Like 2	**0.593 *** **± 0.029**	*0.708* *± 0.037*	*0.678* *± 0.041*
	FTO Fat Mass and Obesity Associated Protein	**0.883 *** **± 0.092**	*0.990* *± 0.072*	*1.041* *± 0.005*
	GTF3C1 General Transcription Factor 3C Subunit 1	**0.909 *** **± 0.038**	*0.946* *± 0.030*	*0.958* *± 0.031*
	LRRC47 Leucine Rich Repeat Containing 47	*0.813* *± 0.172*	**0.807 *** **± 1.059**	*0.954* *± 0.088*
	RBMX RNA Binding Motif Protein X-Linked	**0.803 *** **± 0.064**	*0.913* *± 0.054*	*0.966* *± 0.065*
	SMC3 Structural Maintenance of Chromosomes 3	*0.876* *± 0.081*	**0.854 *** **± 0.059**	*0.841* *± 0.058*
	RANBP3 RAN Binding Protein 3	**1.238 *** **± 0.102**	*1.108* *± 1.069*	*1.036* *± 0.047*
	TOPAZ1 Testis and Ovary Specific PAZ Domain Containing 1	**0.903 *** **± 0.041**	*0.953* *± 0.042*	*0.987* *± 0.042*
Coagulation and immunity	AGT Angiotensinogen	**1.911 *** **± 0.374**	*1.297* *± 0.169*	*1.414* *± 0.067*
	CFI Complement Factor I	**0.868 **** **± 0.049**	*0.946* *± 0.042*	*0.967* *± 0.037*
	F12 Coagulation Factor XII	**0.877 *** **± 0.081**	*0.961* *± 0.065*	*0.984* *± 0.079*
	HP Haptoglobin	*1.059* *± 0.161*	*1.310* *±1.056*	**1.490 *** **± 0.073**
	IGHV3-35 Immunoglobulin Heavy Variable 3–35	**0.711 *** **± 0.069**	**0.761 *** **± 0.049**	**0.675 **** **± 0.055**
	ITIH1 Inter-Alpha-Trypsin Inhibitor Heavy Chain 1	**1.654 *** **± 0.286**	*1.272* *± 2.001*	*1.449* *± 0.069*
	LTF Lactoferrin	*0.855* *± 0.059*	**0.887 *** **± 0.042**	*0.876* *± 0.048*
Vesicle trafficking	AP3B2 Adaptor Related Protein Complex 3 Subunit Beta 2	**0.811 *** **± 0.079**	**0.904 **** **± 0.057**	*0.921* *± 0.072*
	RABEP1 Rabaptin, RAB GTPase Binding Effector Protein 1	**1.371 *** **± 0.146**	*1.116* *± 1.077*	*1.268* *± 0.075*
	SYTL2 Synaptotagmin-Like 2	*1.096* *± 0.071*	**1.155 *** **± 0.048**	**1.186 **** **± 0.077**
Actin cytoskeleton and axon guidance regulation	CLIP1 CAP-Gly Domain Containing Linker Protein 1	**0.834 **** **± 0.045**	*0.942* *± 0.051*	*0.979* *± 0.062*
	DNAAF1 Dynein Axonemal Assembly Factor 1	**0.590 **** **± 0.047**	*0.725* *± 0.053*	*0.683* *± 0.051*
	MAST1 Microtubule Associated Serine/Threonine Kinase 1	*1.213* *± 0.058*	**1.258 *** **± 0.068**	*1.220* *± 0.072*
	MYH9 Myosin Heavy Chain 9	#	**1.717 ****** **± 0.411**	#
	OBSL1 Obscurin Like Cytoskeletal Adaptor 1	*0.954* *± 0.113*	**0.795 *** **± 1.098**	**0.801 *** **± 0.091**
	PLXNC1 Plexin C1	*1.615* *± 0.310*	**2.056 **** **± 3.016**	*1.476* *± 0.051*
	PTPRS Protein Tyrosine Phosphatase Receptor Type S	**0.867 **** **± 0.045**	*0.911* *± 0.028*	**0.826 **** **± 0.034**
	RAI14 Retinoic Acid Induced 14	**0.514 *** **± 0.036**	*0.630* *± 0.030*	**0.598 *** **± 0.037**
Lipoprotein metabolism and transport	APOB Apolipoprotein B	**0.673 *** **± 0.035**	*0.773* *± 0.039*	*0.764* *± 0.037*
	APOH Apolipoprotein H	#	**1.293 **** **± 0.142**	#
Peptide/protein metabolism	DPP3 Dipeptidyl Peptidase 3	**1.319 *** **± 0.152**	*1.107* *± 1.015*	*1.183* *± 0.040*
	PAM Peptidylglycine Alpha-Amidating Monooxygenase	*0.904* *± 0.065*	**0.939 **** **± 0.060**	*0.938* *± 0.054*
	PARG Poly(ADP-Ribose) Glycohydrolase	**1.132 **** **± 0.062**	*1.072* *± 0.055*	*1.021* *± 0.064*
	TG Thyroglobulin	*1.255* *± 0.136*	*1.122* *± 1.070*	**1.246 *** **± 0.011**
Ca^2+^ channel/release	ANKRD36C Ankyrin Repeat Domain 36C	*0.953* *± 0.078*	**0.901 *** **± 0.061**	*0.933* *± 0.042*
	TRPM1 Transient Receptor Potential Cation Channel Subfamily M Member 1	*0.977* *± 0.051*	*0.980* *± 0.041*	**0.928 *** **± 0.030**
Erythrocytes gas exchange	HBA1 Hemoglobin Subunit Alpha 1	**0.546 *** **± 0.035**	*0.675* *± 0.033*	**0.613 *** **± 0.042**
	HBA2 Hemoglobin Subunit Alpha 2	**0.629 *** **± 0.045**	*0.741* *± 0.037*	**0.700 *** **± 0.041**
	HBA2 Hemoglobin Subunit Alpha 2	#	**0.620 *** **± 0.056**	#
	HBB Hemoglobin Subunit Beta	**0.647 *** **± 0.053**	*0.758* *± 0.047*	*0.738* *± 0.044*
	HBB Hemoglobin Subunit Beta	#	**0.597 *** **± 0.033**	#
	HBD Hemoglobin Subunit Delta	**0.605 *** **± 0.042**	*0.744* *± 0.052*	**0.695 *** **± 0.046**
	HBD Hemoglobin Subunit Delta	#	**0.545 **** **± 0.033**	#
	HBE1 Hemoglobin Subunit Epsilon1	**0.604 *** **± 0.037**	*0.749* *± 0.039*	*0.691* *± 0.044*

**Table 2 ijms-26-11922-t002:** Average (±SEM) cumulative number of tender points (out of a potential 30) in the ipsilateral lower limb before (baseline) and during hypertonic saline infusion, with *p* ≤ 0.05 *, *p* ≤ 0.01 **, *p* ≤ 0.001 ***, *p* ≤ 0.0001 ****, and ns: non-significant.

Force	Baseline	Infusion	*p* Value	Significance
5 N	0.0 ± 0.0	0.3 ± 0.3	0.9976	ns
10 N	0.1 ± 0.1	0.8 ± 0.4	0.8440	ns
15 N	0.2 ± 0.1	2.5 ± 0.5	0.0022	**
20 N	0.4 ± 0.2	3.6 ± 0.7	<0.0001	****
25 N	1.3 ± 0.4	5.5 ± 1.0	<0.0001	****
30 N	2.7 ± 0.6	8.7 ± 1.4	<0.0001	****

## Data Availability

The mass spectrometry proteomics data have been deposited to the ProteomeXchange Consortium via the PRIDE [54] partner repository with the dataset identifier PXD070131.

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
