# Peer review of "The Proteome of Acute Muscle Pain: Observations from Acute Hypertonic-Saline-Induced Pain in Humans"

_ijms, 2025, doi:10.3390/ijms262411922_

Round 1
Reviewer 1 Report
Comments and Suggestions for Authors
This study demonstrates systemic molecular alterations resultant from acute HS-induced muscle pain with accompanying sensory interactions. The content is interesting. However, there are many problems that need to be solved.
- The two-cohort design is well thought out, but further clarification on the rationale behind the choice of control conditions would help the reader understand why the specific design was used. Specifically, how do the control conditions relate to the observed results in terms of baseline measurements?
- The cytokine data suggests sustained inflammation, but it would be beneficial to explore the temporal dynamics of cytokine expression in more detail. How does the persistence of cytokine alterations correlate with the duration of muscle pain? Authors showed (TRP) channels and inflammatory response were involved in the development of pain. Here are some references: PMID: 40362205, PMID: 40195186,
- The pain scale used in the study is valid, but more details on how participants subjectively report pain and how these reports were validated would enhance the transparency of the pain measurement process.
- The proteomics data analysis is comprehensive, but the manuscript could benefit from a more detailed explanation of how the identified proteins are selected for further analysis. A more robust description of the bioinformatics tools and algorithms used to analyze the proteomic data would add depth.
- While the statistical analyses are mostly well-conducted, a clearer explanation of why specific tests (e.g., Tukey’s posthoc) were chosen for each comparison would improve the manuscript's methodological transparency.
Partial grammar needs to be modified
Reviewer 2 Report
Comments and Suggestions for Authors
This manuscript presents well-designed and clearly written research. It is methodologically rigorous, and the results are internally consistent. Several revisions, however, are required to strengthen the manuscript and ensure accurate interpretation.
- The study involves short-lived, experimentally induced pain in healthy individuals. The clinical applicability of the findings is therefore limited. Please revise the Abstract, Introduction, and Discussion to frame the conclusions more strictly as mechanistic and hypothesis-generating, rather than implying direct clinical impact.
- Only 13 participants wereanalyzede for proteomics. This affects statistical power. Please: clarify how multiple comparisons were addressed, state whether the study was sufficiently powered for pathway analyses, explicitly acknowledge this limitation in the text.
- Cohort 1 used the forearm flexor carpi ulnaris, whereas Cohort 2 involved the tibialis anterior. These muscles differ in innervation, perfusion, and sensory properties. Please provide a rationale for this design choice or explicitly note it as a limitation.
- Many observed fold changes are modest and could reflect generalized stress responses. Please soften statements suggesting immune dysregulation, oxidative stress, or disrupted homeostasis, and present these as hypotheses rather than definitive conclusions.
- Since no chronic pain population or longitudinal follow-up was included, the manuscript should avoid suggesting relevance to chronic pain development. These links should be reframed as potential areas for future research.
- The absence of contralateral mechanical hyperalgesia indicates that a single acute HS infusion does not induce widespread sensitization within the timeframe studied. Please present this finding cautiously and avoid implying strong conclusions about central sensitization.
- To improve transparency, please include a brief section noting key study limitations such as:
- small sample size,
- acute experimental model,
- different muscles between cohorts,
- potential influence of systemic stress on proteomics.
- Consider adding:
- effect sizes/fold changes for key proteins directly in the Results text,
- brief justification for hypertonic saline concentration,
- clearer indication of which proteins remained significant after correction.
Round 2
Reviewer 1 Report
Comments and Suggestions for Authors
The manuscript has been revised very well. There are still some issues
- the sample size in the proteomics cohort (n=16, with successful venepuncture in 13) is relatively small, which may limit the generalizability of the findings
- The results clearly present the differentially expressed proteins and cytokines during acute muscle pain.The use of heatmaps and tables effectively visualizes the data. Consider adding more specific information on the fold changes and statistical significance of key proteins to strengthen the findings.
Partial grammar needs to be modified
Reviewer 2 Report
Comments and Suggestions for Authors
I endorse this manuscript for publication.
